# Geostress-Adaptive Charge Structure Design and Field Validation for Machinery Room Excavation

**DOI:** 10.3390/s24237738

**Published:** 2024-12-03

**Authors:** Xiaocui Chen, Yuan Mi, Xinru Shuai, Yuan Zheng, Wenhu Zhao

**Affiliations:** 1School of Electrical and Power Engineering, Hohai University, Nanjing 210098, China; xixizu@163.com (X.C.); shuaixinru@hhu.edu.cn (X.S.); zhengyuan@hhu.edu.cn (Y.Z.); 2School of Infrastructure Engineering, Nanchang University, Nanchang 330031, China; whzhao@ncu.edu.cn

**Keywords:** blasting, machinery room, charging structure, rock fragmentation, vibration velocity

## Abstract

The application of blasting in modern engineering construction is prized for its speed, efficiency, and cost-effectiveness. However, the resultant vibrations can have significant adverse effects on surrounding buildings and residents. The challenge of optimizing blasting procedures to satisfy excavation needs while minimizing vibration impacts is a critical concern in blasting excavation. This research addresses this challenge through the development of a 3D simulation and analysis model for an underground pumped storage power plant in East China, utilizing the LS-DYNA finite element analysis software. To explore the influence of charging structures on rock fragmentation and vibration propagation, three distinct blasting programs were formulated, each featuring varied configurations within the machinery room. The analysis revealed that the adoption of an optimized charging structure can significantly decrease damage to the protective layer by approximately 40%, while also reducing the impact on the upstream and downstream side walls by 27.25% and 12.03%, respectively, without compromising the efficacy of the main blast zone. Moreover, the vibration velocities at the remote measurement point were found to be reduced across multiple directions, indicating effective control of the vibration effects. The post-implementation of the optimized blasting strategy at the site, the assessment of the retained surrounding rock integrity, and the impact on protected structures demonstrated that the proposed solution met satisfactory outcomes. This study underscores the potential of simulation-based optimization in managing vibration risks during blasting operations, offering a valuable tool for engineers and practitioners in the field of underground construction.

## 1. Introduction

The implementation of China’s dual-carbon policy has precipitated a surge in the development of extensive water conservancy and hydropower initiatives, including the Xiluodu, Xiaowan, Waterfall Gully, Wudongde, and Baihetan projects [1]. Given the varied topographical and geomorphological characteristics across different regions, these infrastructure projects necessitate the execution of extensive, high-intensity rock excavation for the construction of underground cavern complexes. It is anticipated that, in the immediate future, blasting will continue to be the predominant technique employed for rock tunnel excavation [2,3,4,5,6,7,8]. Consequently, the numerical modeling of drilling and blasting excavation-induced damage, coupled with the optimization of parameter design, stands as a critically significant area of research within the disciplines of water conservancy engineering and civil engineering [9,10,11,12,13].

To achieve the rational optimization of blasting parameters with the aim of cost reduction while guaranteeing the efficacy and stability of the excavation process and the surrounding rock mass, a substantial body of research has been conducted by scholars, both domestically and internationally. Empirical evidence indicates that the blast-induced stress waves and the resultant explosive gasses are the primary factors precipitating rock fragmentation [14,15,16]. The impact of blast stress waves serves to initiate the formation of initial cracks. Subsequently, the quasi-static pressure exerted by the expanding blast gasses contributes to the propagation and further expansion of these nascent fractures [17]. Substantial field trials have yielded evidence indicating that the adoption of smooth blasting techniques and the application of pre-splitting blasting methodologies are efficacious in the management of blasting-induced damage effects [18,19,20,21]. In addition, there are many damage control techniques. Park et al [22] contends that the adoption of the line drilling technique is conducive to diminishing the detrimental effects on the preserved rock mass, as it is hypothesized to act as a barrier that intercepts and absorbs the propagating waves emanating from blast-generated forces. Similarly, the equivalent protective effect can also be realized through the strategic creation of notched holes, lined with appropriate shielding materials, within the retained rock mass [23,24]. Nevertheless, the aforementioned strategies for blast damage mitigation are predicated on modifications to the drilling parameters associated with the position of the blasting contour. The boreholes positioned within these specific locations typically exhibit smaller diameters, inter-bore spacings, and inter-row spacings [25]. In the case of underground caverns within pumped storage power stations, the application of these smaller drilling parameters is confined to the peripheral holes. For large-scale layered excavations within the machinery room, the drilling parameters are generally much larger, often falling short of the stringent requirements for contour blasting. As such, the manipulation of blasting parameters stands as the predominant factor in the effective control of rock mass damage.

The charge structure emerges as a pivotal determinant of the efficacy of blasting operations, ranking prominently among the core research focal points within the discipline of blasting technology [26,27,28,29]. The axial spacing charging structure, a prevalent configuration in blasting engineering, is characterized by its ability to decouple the various mechanical effects of the blast along the axis. This attribute, on one hand, stems from its inherent axial uncoupling coefficient, which mitigates the interplay of these effects. On the other hand, the differential axial spacing between charges, influenced by the distinct physical and mechanical properties of the intervening medium, yields varying degrees of impact on the borehole wall and energy dissipation patterns.

The above studies indicate a complexity in the application and resultant rock fracturing efficacy of spaced charging configurations within underground cavern blasting operations. To delve into this complexity, this study employs the explicit finite element analysis software LS-DYNA R8.1 to investigate the fragmentation rates and vibration responses of a modeled machinery room subjected to various charging scenarios. An optimal charging scheme is identified through this simulation process and subsequently field-applied to assess the blasting outcomes. Concurrently, field-measured values are juxtaposed against the numerical simulation results to ascertain the accuracy and rationality of the selected charging scheme, thereby validating its efficacy in real-world applications.

## 2. Project Introduction and Field Test

### 2.1. Project Introduction

The excavation of the primary plant cavern encompasses a multifaceted architectural layout, which includes the installation room, machinery room, auxiliary powerhouse, and the two ventilation rooms. The left ventilation room is integrated with a drainage gallery, which spans a total length of 16 m, facilitating the effective management of water drainage. The installation room, machinery room, and auxiliary powerhouse measure 46 m, 111.5 m, and 19.5 m in length, respectively. The right ventilation room is coupled to a safety tunnel, measuring 12 m in length, which serves as a critical interface for air circulation and emergency access. The entire structure exhibits symmetry about the central axis of the cavern. The main transformer room is positioned on the downstream side of the main plant cavern. It is oriented parallel to the main plant cavern, maintaining a net distance of 40 m from the opening of the main plant. The comprehensive architectural model is depicted in Figure 1a.

In the context of the phased excavation of an underground power plant from the top down, the surrounding rock experiences significant damage, with substantial volumes of plastic dissipation energy being released. Concurrently, substantial displacements occur around the excavation chambers, and the redistribution of surrounding rock stresses ensues, contributing to a degradation in the stability of the rock mass. To address these challenges and mitigate the stress disturbances during the construction phase, a supportive intervention strategy is implemented following the excavation of the machinery room. This entails the installation of three rows of anchor rods along the rock-anchored beams. The upper two rows are composed of Φ36 mm steel bars, which are anchored into the rock mass at a depth of 10.2 m. The lower row consists of Φ28 mm steel bars, anchored at a depth of 7.8 m. The entire array of anchor rods is arranged in a prune-like configuration. Figure 1b illustrates the localized support structure during the construction phase.

### 2.2. Field Test

#### 2.2.1. Maximum Vibration Velocity

To facilitate the quantification of vibration effects that are resultant from the blasting operation within the targeted excavation section, an intricate network of measurement nodes was deployed at varying depths along the rock-anchored beam. The placement of these nodes adhered to the principle of “near dense and far sparse”, a strategic distribution scheme designed to enhance the spatial resolution of the monitoring system while minimizing the number of sensors required. The repercussions of explosive phenomena impart a discernible vibrational perturbation to the particle affixed to the sensor. This resultant vibration induces a relative displacement within the sensor’s integral magnetic subsystem and its associated coil. Consequently, this mechanical movement is transduced into an electric potential, which is then relayed through a CPU-mediated interface to a computational platform for subsequent analysis. In light of these dynamic interactions, the 6850 Network Vibrometer has been selected as a field-deployable instrument for the precise measurement of peak vibrational velocities. To secure the sensors to the rock-anchored beam, gypsum was utilized at each measuring point, providing the robust and stable connection necessary for accurate data acquisition. The selected instrumentation and the on-site sensor layout are visually depicted in Figure 2.

#### 2.2.2. Wave Speed and Anchor Stress

To mitigate the extensive damage to the retained rock caused by the detonation process, the 0 + 000.5 and 0 + 009.5 sections on the right side of the machinery room were designated for the assessment of relaxation depth, as shown in Figure 3a. Subsequently, 6 m deep boreholes were meticulously drilled along the upstream and downstream side walls, identified as S1 and S2. The evaluation was conducted utilizing the single-hole acoustic method, which entailed a point-by-point testing regimen from the base to the summit of the boreholes. The spacing between consecutive test points was maintained at 0.2 m, with depth corrections being applied after every five points were tested. The relaxation depth of the surrounding rock was then determined by analyzing the alterations in wave velocity measurements, providing a quantitative basis for understanding the rock mass’s response to the excavation stresses. Furthermore, as illustrated in Figure 3b, to augment the monitoring of structural stability, stress gauges were integrated within the rock-anchored beam. These gauges were designed to monitor the stress distribution across each row of anchor rods post-blasting.

## 3. Numerical Model

### 3.1. Overall Model

Pursuant to the spatial configuration of the power plant area, a representative set of working conditions pertinent to the construction of the machinery room within the underground power plant was chosen for blasting simulation analysis. Specifically, the selected scenario encompassed the range from 0 + 000.000 to 0 + 040.000 m along the right side of the plant and from 187.3000 to 237.300 m above the plant elevation. Given the absence of significant geological faults in the engineering vicinity and the relatively minor influence of rock joints on blasting-induced damage, the simulation model was designed without incorporating discontinuities within the rock mass. The constructed simulation model is depicted in Figure 4.

### 3.2. Material Model

#### 3.2.1. Rock Mass

A preliminary geological survey and on-site analysis have revealed that the surrounding rock formations within the project vicinity are predominantly composed of slightly weathered quartz monzonitic porphyry, a classification of hard rock material, as illustrated in Figure 5. To elucidate the mechanical characteristics of the rocks within the engineering zone, laboratory testing was conducted on samples of the primary exposed quartz monzonitic porphyry. The average values of these measurements were utilized to establish a baseline for the rock’s mechanical properties. The rock density *ρ*_0_ was determined to be 2730 g kg/m^3^, the uniaxial compressive strength *f*_c_ was measured at 169 MPa, and the elastic shear modulus *G* was found to be 20 GPa. In light of the simulation’s requirement to capture the dynamic behavior of rock material subjected to high strain rates and significant deformations, the Riedel–Hardin–Tucker (RHT) model was adopted for its capacity to rigorously determine the elastic yield surface, the failure surface, and the residual strength surface. The statistical data for each parameter associated with this model are presented in Table 1.

#### 3.2.2. Explosive and Air

In order to replicate the behavior of TNT emulsion explosives, as employed in practical applications, the *MAT_HIGH_EXPLOSIVEOSIVE_BURN material model within the LS-DYNA software was employed. This model was configured with a density of 1631 kg/m^3^, a detonation velocity of 6930 m/s, and a detonation pressure of 21 GPa, parameters that accurately reflect the physical properties of the explosive. The temporal evolution of the volume, pressure, and energy profiles of the explosive products during the detonation process were delineated through the application of the Jones–Wilkins–Lee (JWL) equation of state, a well-established thermodynamic model that is formally expressed as follows:(1)$$P=A\left(1-\frac{w}{R_{1}V}\right)e^{-R_{1}V}+B\left(1-\frac{w}{R_{2}V}\right)e^{-R_{2}V}+\frac{WE_{0}}{V}$$
where *P* is the detonation pressure; *A*, *B*, *R*_1_, *R*_2_, and *w* are the detonation parameters of explosives; *V* is the relative volume of the blast products; and *E*_0_ is the initial specific internal energy. For the current simulation, the following values have been adopted: *A* = 371.2 GPa, *B* = 3.2306 GPa, *R*_1_ = 4.5, *R*_2_ = 0.95, *W* = 0.3, *E*_0_ = 7 GPa, and *V* = 1.0.

Air is modeled through the employment of Material Model #9, designated as *MAT_NULL. The density of this air medium has been assigned a value of 1.2 kg/m^3^. The thermodynamic behavior of air is governed by a linear polynomial equation of state, identified as *EOS_LINEAR_POLYNOMIAL. This equation of state delineates the interdependence of pressure, density, and internal energy, and is mathematically formulated as follows:(2)$$P=C_{0}+C_{1}\mu +C_{2}\mu^{2}+C_{3}\mu^{3}+\left(C_{4}+C_{5}\mu +C_{6}\mu^{2}\right)E$$
where *E*_0_ is the internal energy per unit volume; $\mu =\frac{\rho }{\rho_{0}}-1$, $\frac{\rho }{\rho_{0}}$ is the ratio of the current fluid density to the initial fluid density; and *C*_1_~*C*_6_ are the parameters related to the equation. The values taken in this simulation are as follows: *C*_0_ = *C*_1_ = *C*_2_ = *C*_3_ = *C*_6_ = 0, *C*_4_ = *C*_5_ = 0.4, *V*_0_ = 0, and *E*_0_ = 0.25 MPa.

#### 3.2.3. Gun Mud and Anchor Rod

For the purpose of simulating the obstructive material within each borehole, the *MAT_SOIL_AND_FOAM material model was adopted. This model was selected due to its capacity to replicate the mechanical properties of the filling material, and it was configured with a density of 1800 kg/m^3^ and a shear modulus of 63.85 MPa, values that correspond to the empirical characteristics of the gun mud composition.

The anchor rod was defined through the application of the *MAT_PLASTIC_KINEMATIC elastic–plastic material model. This model was parameterized to reflect the anchor rod’s material properties, with a density of 7850 kg/m^3^, an elastic modulus of 210 GPa, a Poisson’s ratio of 0.3, and a yield stress of 350 MPa. These values were determined based on the mechanical characteristics of the anchor rod material, ensuring that the simulation adequately captures the material’s response to loading conditions.

### 3.3. Related Settings

To mitigate the potential impact of reflected waves at artificial boundaries on the simulated rock structure, non-reflective boundary conditions were imposed on all six model surfaces, with the exception of the free surface. This configuration was adopted to mimic the infinite extent of the rock block within the computational domain. In light of the rock settlement observed in the engineering area and the stress relaxation phenomena encountered during excavation, the numerical model was designed to incorporate the effect of geostress. An average stress value of 9 MPa was selected based on empirical test data to reflect the in situ conditions accurately.

In the simulation, the rock-anchored beam was modeled in contact with the surrounding rock, with static and dynamic friction coefficients set to 0.2 to account for the interaction between the beam and the rock mass. To capture the mechanical behavior of the rock under complex loading conditions, custom functions were set up to simulate damage. This function enables the depiction of tensile stress-induced strip-shaped cracks and shear strain-induced cylindrical crushing zones, thereby providing a detailed representation of the rock’s response to tensile and shear forces.

Furthermore, to ensure the proper integration of fluid elements with solid elements and to manage interactions between solid elements and beam elements, sophisticated algorithms such as penalty coupling and motion constraints were implemented. These measures are crucial for achieving a cohesive and accurate simulation of the coupled mechanical behavior of the rock mass, rock-anchored beam, and any fluid dynamics involved in the analysis.

## 4. Charge Structure Optimization

### 4.1. Model Simplification

To delve into the optimization of the charging method within the midsection groove, with the aim of increasing the fragmentation rate and diminishing the vibrational impact of blasting without incurring excessive economic costs, a comprehensive numerical analysis was conducted on a simplified model derived from the overall configuration depicted in Figure 6a. This simplified model, presented in Figure 6b, encompasses dimensions of 50 m by 40 m by 6 m and incorporates monitoring points set at 5 m intervals along the central line from 10 m to 50 m to capture vibration velocity data across various orientations under diverse operational conditions. The primary blasting area is delineated in Figure 6c. The borehole diameter was chosen to be 210 mm, with an inter-hole spacing and row spacing of 3.6 m. The depth of the boreholes was determined in alignment with the height of the excavation section, which was established at 6 m. The significance of the decoupling coefficient of the explosive charge in mitigating energy loss and enhancing explosive efficiency was recognized. Consequently, a decoupling coefficient of α = 1.28 was selected to optimize the performance of the explosive charge [30]. Drawing upon practical engineering experience, it was acknowledged that the surrounding rock mass adjacent to the blast holes would experience considerable influence during the blasting event. To facilitate detailed monitoring of the deformation and failure dynamics of this critical region, a local mesh densification strategy was implemented. The simulation model for this explosion incorporates a meticulous mesh structure, with a total of 824,913 computational nodes and 7,830,009 hexahedral elements. The detailed schematic of the grid segmentation and an amplified visualization of the blast hole are presented in Figure 6d. Given the extensive memory and computational time requirements, a server with an AMD Ryzen Threadripper 3970X processor (Advanced Micro Devices, Inc., Santa Clara, CA, USA) was employed for the simulation. The system utilized featured 32 processing cores and was configured with a memory allocation of 256 GB to accommodate the model’s execution.

### 4.2. Blasting Plan

Following the establishment of the optimal parameters for hole depth, aperture, and charge diameter, a detailed blasting analysis was conducted on the machinery room of underground powerhouse. This analysis was performed within the context of three distinct charging configurations, each presented in Figure 7. Each individual section of explosive or gun mud utilized in these configurations was uniformly sized at 0.5 m in length. Case A features a single, continuous section of explosive, which spans a total length of 3 m. In contrast, Case B employs two separate sections of explosive, each measuring 1.5 m in length, resulting in a cumulative length of 3 m. Case C, on the other hand, distributes the explosive into six smaller sections, each 0.5 m in length, thereby maintaining the same total length of 3 m as the other plans.

### 4.3. Rock Damage

To validate the blasting efficacy under diverse charging arrangements, the damage cloud diagrams at 0.6 ms, 1.5 ms, and 30 ms post-detonation were selected for analysis, as depicted in Figure 8. These damage cloud diagrams depict the damage degree of the rock mass, where *D* = 1 signifies the complete damage to the surrounding rock, with severe crushing, and *D* = 0 indicates no damage, with the surrounding rock properties being unaffected by the blasting-induced loads. Observations indicate that the initiation of explosive detonation within each borehole is accompanied by the pronounced fragmentation of the surrounding rock mass. This is due to the reflection of stress waves at the borehole wall, where the reflected tensile stress waves cause extensive damage to the rock matrix. Over time, the cumulative impact of blasting results in the formation of a crush zone that radiates outward from the blast center. Concurrently, the rock damage around the internal blast hole exhibits radial fracturing characteristics. This is because the cylindrical symmetry of stress wave propagation is a fundamental characteristic of explosive detonation, regardless of the initiation pattern. As the stress waves emanate from the porous explosive structure, they undergo superposition and propagate spherically towards the surrounding boundaries. This spherical wavefront distribution leads to a slightly enhanced fragmentation effect along the centerline of each borehole compared to adjacent regions. Finally, the expansion of the initial radial fractures is accelerated by the influence of high-pressure explosive gasses, leading to their rapid stretching and widening. Therefore, the selection of an appropriate hole and row spacing is critical. When such conditions are met, the fissures generated by individual boreholes interconnect, forming a cohesive network. This interconnected fissure system undergoes further expansion and extension, ultimately resulting in the disintegration of the rock mass into smaller, discrete fragments.

To facilitate a more detailed examination of rock fragmentation, units exhibiting damage values exceeding 0.6 post-blasting were classified as failed and were subsequently excluded from analysis. The residual rock units were monitored and are depicted in Figure 9. The rock morphology resulting from blasting with schemes A, B, and C is denoted by purple, orange, and yellow, respectively. The results demonstrate significant discrepancies in rock fracturing among the three scenarios. Case A, which featured a singular section of concentrated explosives, exhibited a pronounced superposition of stress waves. The coupled stresses exerted strong lateral impacts on the protective layers on both sides of the blasting zone, leading to substantial damage in these areas. For the two sections of the charge (Case B), the main blasting area experienced damage expansion and penetration, enabling the successful propagation of cracks. Conversely, the six-section charge (Case C) did not appear to achieve the desired blasting excavation effect, suggesting that the rock mass may not have been effectively fragmented. Notably, all three schemes exhibited extensive damage at the free surface of the intermediate section. This observation may be attributed to the incomplete absorption of stress waves, which allows reflected tensile stress waves to propagate back into the rock mass, leading to subsequent damage.

To enhance the quantification of rock fragmentation, the proportion of fragmented rock units within the midsection groove, as well as in the upstream and downstream protective layers, was assessed across various charge structures. The outcomes of this analysis are presented in Figure 10. As depicted in the figure, Case A exhibits the most pronounced rock breaking effect at the midsection groove, achieving fragmentation exceeding 0.5. However, this efficacy is accompanied by a higher level of damage to the upstream and downstream protective layers, with respective damage fractions of 0.4385 and 0.3919. In contrast, Case B achieves a rock breakage rate of 0.4966 at the midsection groove, a decrease of 7.26% from Case A, yet incurs significantly less damage to the protective layers, with damage fractions of 0.2657 and 0.2359 for the upstream and downstream layers, respectively. The reduction in damage for the upstream protection layer in Case B is 39.81% compared to Case A, and for the downstream protection layer, it is 39.41% lower. Case C demonstrates a less effective fragmentation due to the dispersal of energy release, resulting in a damage rate of only 0.4089 in the midsection groove and 0.1765 and 0.1272 for the upstream and downstream protection layers, respectively. The aforementioned data further underscore that an optimized charging program can enhance the fragmentation within the primary blast zone without an incremental requirement for explosives. It is evident that Case A and Case B outperform Case C in terms of rock fragmentation efficiency.

In the context of this project, it is imperative to guarantee that the foundation surface maintains adequate bearing capacity and exhibits robust stability, thereby safeguarding the operational reliability of the hydraulic infrastructure. To this end, the impact resulting from blasting activities within the midsection groove, particularly on the side walls, necessitates heightened attention. The utilization of MATLAB R2022a software was employed to execute grayscale conversion and subsequent binarization on the imagery depicting the peripheral rock damage at the interface between the protective layer and the sidewalls, both at the upstream and downstream contact surfaces. Figure 11 illustrates the sidewall damage resulting from the selected charge structure B. Notably, the downstream sidewall exhibits a more pronounced degree of damage when compared to its upstream counterpart. The observed damage patterns are characterized by a centrosymmetric morphology. The analysis of the damage area, as quantified by regions exhibiting damage values exceeding 0.6, is presented in Table 2. The data reveal that the charge structure A precipitated the most extensive damage to the side wall surface, with the upstream side wall sustaining 37.45% more damage than that caused by structure B and the downstream side wall experiencing 13.68% greater damage. Conversely, charging structure C exerted the least influence on the upper and lower sidewalls, resulting in damage levels of only 0.3223 and 0.4927, respectively.

### 4.4. Vibration Response

Employing the explicit finite element analysis software LS-DYNA, the blasting was simulated over a total duration of 30 milliseconds, with the simulation divided into 100 time steps, each corresponding to a time increment of 0.03 milliseconds. The temporal evolution of vibration velocity across various measurement points was meticulously tracked, with the resulting time curves depicted in Figure 12. In the context of the spatial distribution of measuring points 1 to 6, which extend from the vicinity to the farthest distance from the blast center, a temporal delay in the manifestation of peak vibration velocities was observed at the rear points relative to the front. Specifically, the peak vibration velocity at measuring point 6 was observed to occur approximately 5 milliseconds later than at measuring point 1. In addition, the cumulative effects of the stress waves emanating from multiple boreholes, including superposition, reflection, and refraction, result in a more dispersed vibration energy distribution at the rear measuring points. This dispersion leads to the generation of multiple sets of vibration velocity peaks. Nonetheless, the intense vibrations are largely subsided within a 10-millisecond timeframe. Notably, the peak vibration velocities recorded under each charging scheme exhibited a decline with increasing distance from the blast center, with the highest value recorded at the closest measuring point, 1. Case A yielded the highest peak vibration velocity, reaching 3.34 m/s. Case C, which demonstrated a peak vibration velocity of 3.03 m/s, was found to lie intermediate between the velocities of Cases A and B. Case B exhibited the lowest peak vibration velocity, measuring 2.91 m/s. This value is 12.87% less than that of Case A and 3.96% less than that of Case C.

To refine the selection of the optimal charge structure, a comparative analysis was conducted on the peak vibration velocities at each measurement point, considering both the horizontal tangential and gravitational directions, as presented in Figure 13. The results revealed that across the three investigated scenarios, the peak velocities recorded in the horizontal tangential direction consistently exceeded those observed in the gravitational direction, and the variation in the peak vibration velocity in both directions showed a consistent trend. Notably, the peak vibration velocity for Case A was found to be superior to that of Case B and Case C at measurement points 1 and 2. Conversely, at measurement point 3, Case C demonstrated a higher peak vibration velocity than both Case A and Case C. At the measurement points 4, 5, and 6, the peak velocities were found to be relatively similar across all schemes. Specifically, at measurement point 1, the peak vibration velocity in the horizontal tangential direction for Case A was determined to be 2.87 m/s, marking a 13.89% increase when compared to Case B. In the gravitational direction, the peak velocity for Case A was 1.71 m/s, corresponding to a 17.12% enhancement over that of Case B. At measurement point 3, the peak velocity in the horizontal tangential direction for Case C was recorded at 1.18 m/s, representing a 12.38% increase relative to Case B. The peak velocity in the gravitational direction for Case C was 0.60 m/s, which is 15.38% higher than the corresponding velocity of Case B. These findings indicate that the selection of Case B can effectively mitigate the vibration response to a certain extent.

## 5. Field Application

Due to the inability of the charge structure A to ensure a flat and smooth contour surface of the machinery room, the charge structure C fails to achieve the anticipated blasting excavation effects, while the charge structure B, in satisfying the blasting requirements, generates the minimum vibration response. Therefore, it is deemed that charge structure B is the optimal solution. The optimal charging scheme identified for field deployment is depicted in Figure 14, and the overall model depicted in Figure 4 was chosen for the corresponding numerical simulation. Post-blasting, wave velocities were quantified in the side wall holes positioned at 0 + 000.5 (Section 1) and 0 + 009.5 (Section 2) along the right side of the plant. The variation in these wave velocities, with respect to hole depth, was plotted, as illustrated in Figure 15. The analysis of the wave velocity trends at Section 1 and Section 2 indicated a notable consistency. The average wave velocities recorded from the orifice to a depth of 0.4 m were relatively small. Specifically, at Section 1, the mean wave velocity was measured to be 4547 m/s in the upstream and 4317 m/s in the downstream. Similarly, at Section 2, the average wave velocity recorded was 4590 m/s upstream and 4334 m/s downstream. The wave velocity recorded from a depth of 0.4 m to the depth of the borehole’s floor exhibits a relatively high and stable profile, and the minimum, average, and maximum wave velocities recorded at various locations are comprehensively presented in Table 3. The analysis of the wave velocity profile revealed a distinct inflection point at a depth of 0.4 m. Characteristically, the sound wave above this inflection point is smaller, whereas below the inflection point is elevated and exhibits stability within a specified range. The transition between these two phases is characterized by a change rate of approximately 10–20%, indicating a relaxation depth of 0.4 m. In comparison to analogous projects under similar geological conditions, the determined relaxation depth was found to be notably smaller, categorizing it as a case of overall slight relaxation with localized medium relaxation [31,32]. This observation suggests that the surrounding rock maintains a high degree of overall stability subsequent to blasting. It is evident that a well-chosen charging scheme can mitigate the intense impact forces generated by blasting, thereby minimizing the potential damage to the surrounding rock strata.

Given the substantial influence of blasting and excavation activities on the integrity of the rock-anchored beam within the machinery room, it was imperative to mitigate the risk of significant cracking and deformation. To this end, the vibration velocity at each monitoring point and the stress on the anchor rod would be meticulously recorded. Drawing upon the lessons learned from analogous projects and the findings of the site survey, it was observed that the rock-anchored beam experienced the highest vibration speed in the direction aligned with the gravitational force. Consequently, the vibration velocity in this direction was selected for comparison with the numerical simulation values, as depicted in Figure 16a. Subsequent to the blasting operation, the numerical simulation results of the stress distribution on the upper row of anchors are presented in Figure 16b.

Upon examining Figure 16a, it is apparent that the zenith of vibration velocity is situated within the 7 to 11 m range. The peak vibration velocity recorded in situ was measured to be 11.46 cm/s. Upon calculating the discrepancy between the measured values and those derived from numerical simulations, it was ascertained that these differences are uniformly within the 15% tolerance range. In alignment with the “Blasting Safety Regulations” [33], the permissible particle vibration velocity for concrete of 7–28 days in age is specified to be between 7 and 12 cm/s. Consequently, the conducted blasting activities adhered to these safety standards. Figure 16b reveals that the majority of the upper row anchors are subjected to tension, with a notable trend of stress increasing with depth before decreasing. The localized axial stress exhibited a maximum value of 34.78 MPa, which, while slightly exceeding the field-measured value of 33.46 MPa, remains within the acceptable range as per safety guidelines. These findings indicate that both the measured vibration velocities and anchor stresses at the rock-anchored beam are not only less than the prescribed safety thresholds but also demonstrate a high degree of correspondence with the numerical simulation results. This substantiates the accuracy and rationality of the selected design approach.

## 6. Conclusions

This investigation proposes a charging strategy designed to mitigate the vibration effects on the distal rock mass while regulating the fragmentation rate achieved through blasting. To ascertain the optimal charging structure for the machinery room of a pumped storage power plant, a calibrated numerical model was employed to conduct a comprehensive analysis of various detonation scenarios for the excavation section within the machinery room. Through the statistical analysis of rock fragmentation at diverse locations and the vibration responses of the surrounding rock, the following outcomes were observed:(a)Case A demonstrated the most effective rock-breaking performance at the midsection groove, achieving a fragmentation rate exceeding 0.5. However, this was accompanied by significant damage to the upstream and downstream protective layers, with damage fractions of 0.4385 and 0.3919, respectively. Case B, while yielding a slightly lower fragmentation rate of 0.4966 at the midsection groove compared to Case A, resulted in significantly reduced damage to the protective layers, with decreases of 39.81% and 39.41% for the upstream and downstream, respectively. Case C exhibited a lower fragmentation rate of 0.4089 at the midsection groove, with the damage distribution being discontinuous, which is attributed to the dispersed energy release.(b)The damage areas resulting from the three schemes at the side walls were all centrally symmetric. Case A imparted the most substantial impact on the upstream and downstream peripheral rock side walls, with damage areas accounting for 0.5278 and 0.6142. Case C produced the smallest damage area, at 0.3224 and 0.4927. Case B had damage areas slightly larger than Case C but exhibited a 27.25% reduction in upstream damage and a 12.03% reduction in downstream damage compared to Case A. Given the necessity to maintain a flat and smooth contour surface within the machinery room and to minimize damage to the retained rock mass, Case A was deemed unsuitable for practical application.(c)Vibration measurements indicated that Case A produced the highest peak vibration velocity, reaching 3.34 m/s. Case C’s peak vibration velocity fell between those of Cases A and B, at 3.03 m/s, while Case B exhibited the lowest peak vibration velocity, at 2.91 m/s, which was 12.87% less than that of Case A and 3.96% less than that of Case C. Furthermore, a comparative evaluation of peak vibration velocities in the horizontal tangential direction was conducted against those in the gravitational direction. Notably, the observed trends in peak velocity were consistent across both directions. Regardless of the directionality of measurement, Case A demonstrated higher peak velocities compared to both Case B and Case C at measurement points 1 and 2. Conversely, at measurement point 3, Case C exhibited peak velocities that were greater than those of both Case A and Case C. At measurement points 4, 5, and 6, the peak vibration velocities of the three schemes were essentially equal.

Upon a thorough evaluation of three distinct charging schemes, Case B was identified as the superior strategy and was subsequently implemented in the excavation section within the machinery room. Field-based assessments were conducted to summarize and analyze the resultant damage to the surrounding rock, as well as the vibration velocity and tensile stress at the rock-anchored beams. The results indicate that Case B not only satisfies the requirements for the desired range of rock fracturing, but also ensures that the vibration velocity and tensile stress levels at the rock anchor beams remain within the safe operational limits. These values are also found to be in close agreement with those predicted by numerical simulations. The proposed scheme effectively regulates the rate of rock fracturing while concurrently minimizing the vibration impact, thereby demonstrating significant utility in similar underground cavern group blasting operations. It should be noted that the established model is applicable only to areas with moderate to low ground stress. The rock mass size of the segment to be excavated in the project can be proportionally converted to the corresponding dimensions aforementioned. Satisfactory blasting schemes can be achieved without incurring any additional economic costs.

## Figures and Tables

**Figure 1 sensors-24-07738-f001:**
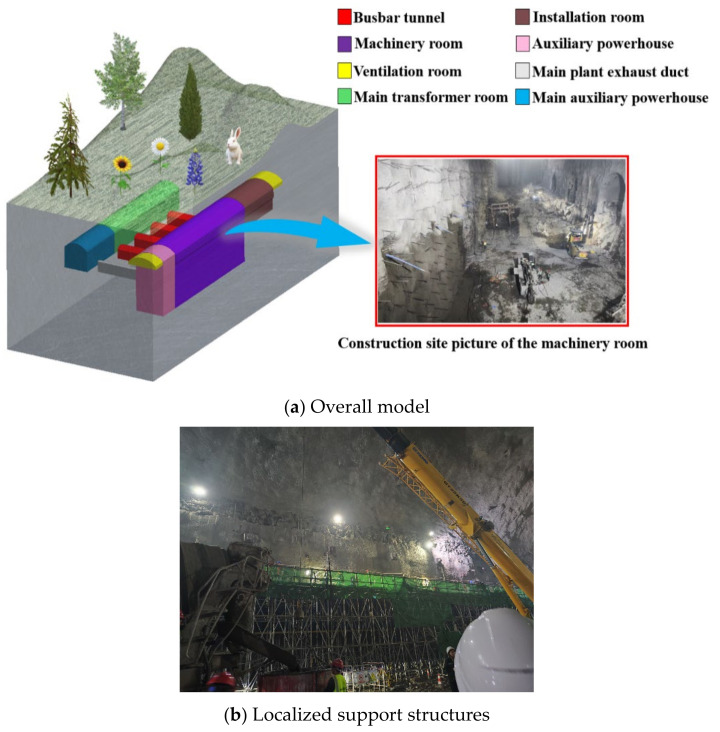
Schematic diagram of the cavern of the underground factory building.

**Figure 2 sensors-24-07738-f002:**
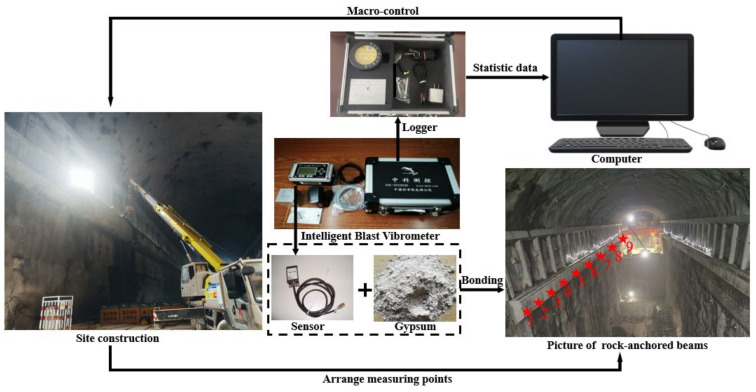
Schematic diagram of vibration velocity measurement.

**Figure 3 sensors-24-07738-f003:**
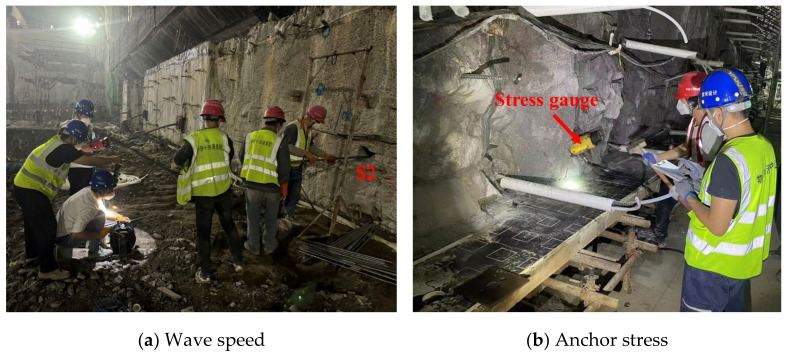
Field photographs of wave speed and stress.

**Figure 4 sensors-24-07738-f004:**
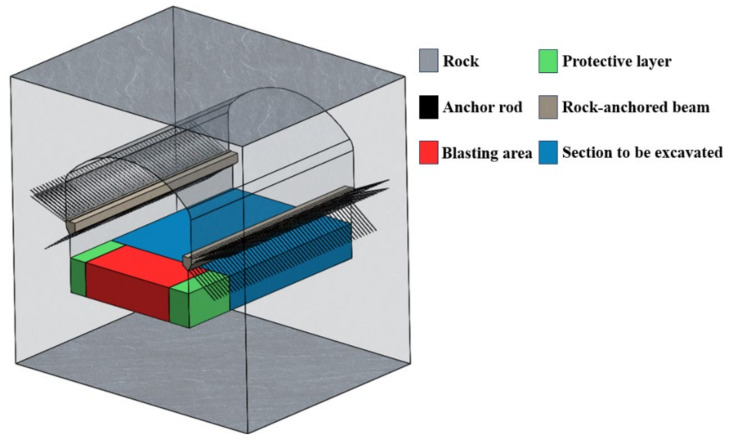
Overall model of the plant section to be excavated.

**Figure 5 sensors-24-07738-f005:**
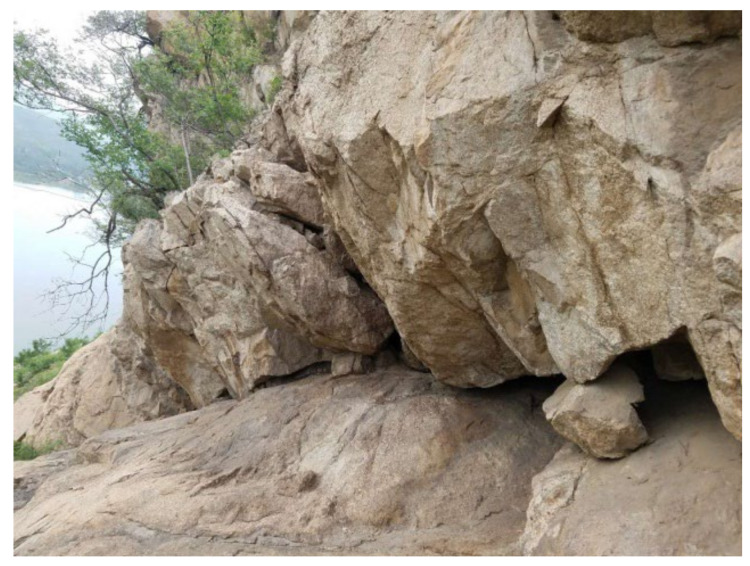
Localized exposed bedrock of a pumped storage power plant.

**Figure 6 sensors-24-07738-f006:**
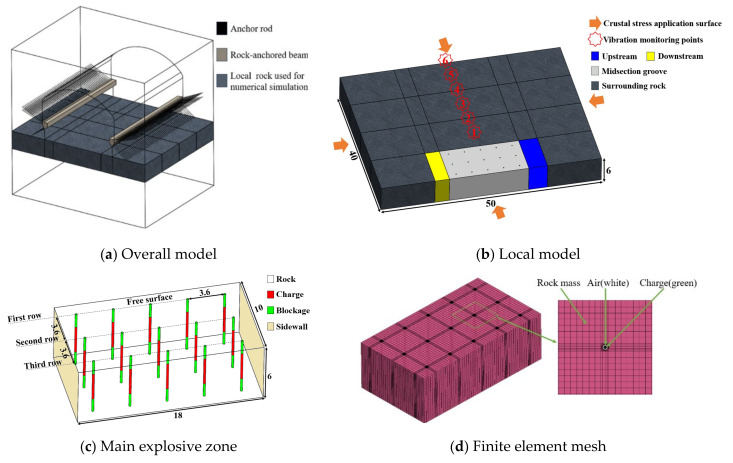
Schematic diagram of the excavation section model and grid segmentation in the machinery room.

**Figure 7 sensors-24-07738-f007:**
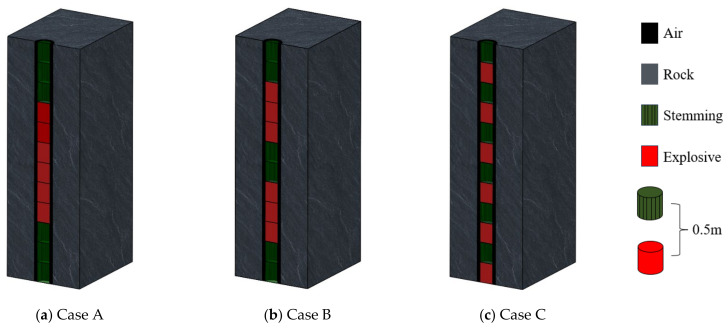
Three charge structures.

**Figure 8 sensors-24-07738-f008:**
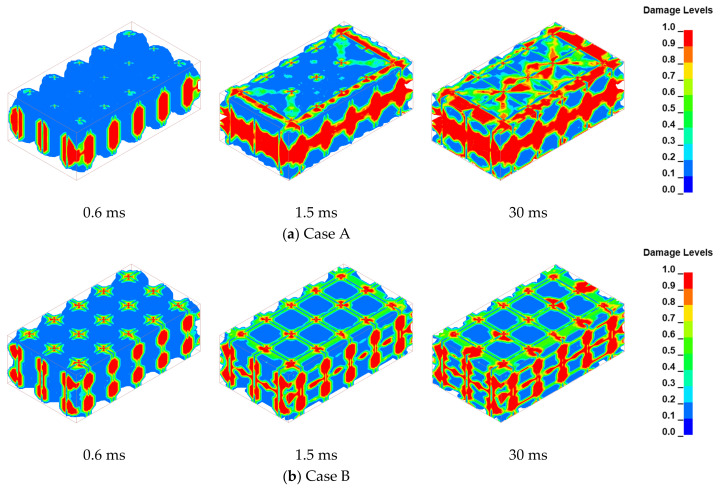
Damage evolution of the surrounding rock in the blasting area.

**Figure 9 sensors-24-07738-f009:**
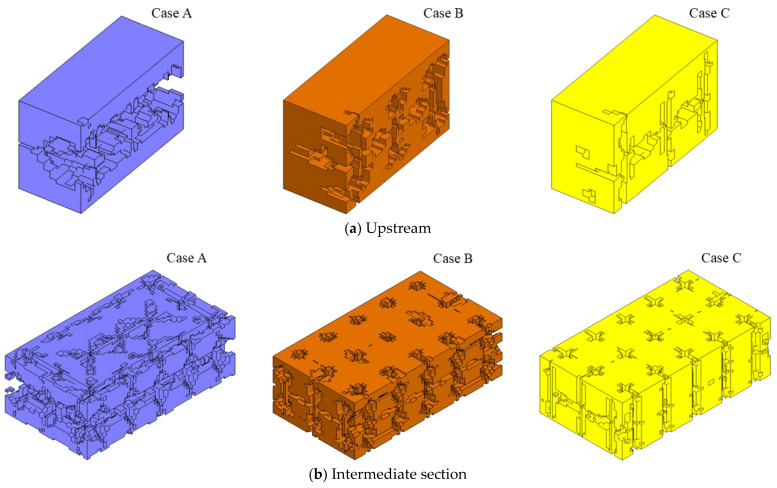
Rock fragmentation morphology of the excavated section of the machinery room at 30 ms.

**Figure 10 sensors-24-07738-f010:**
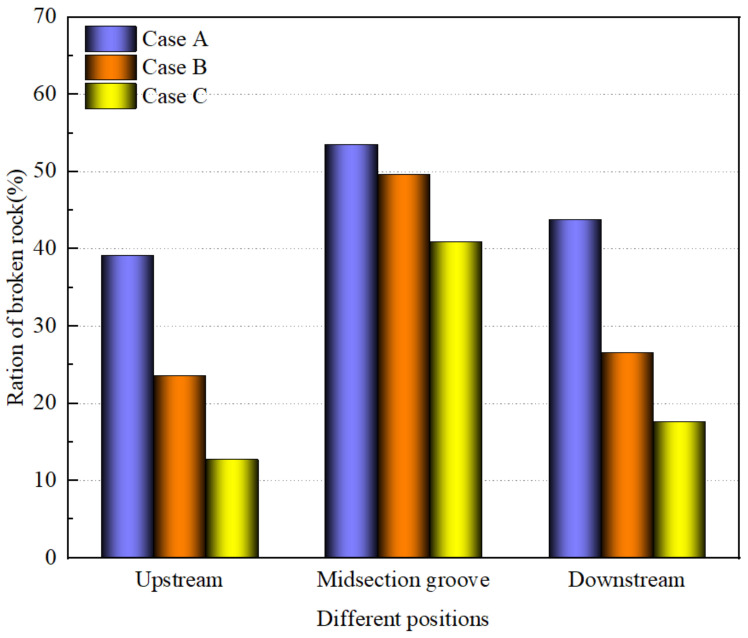
Proportion of rock fragmentation at different locations.

**Figure 11 sensors-24-07738-f011:**
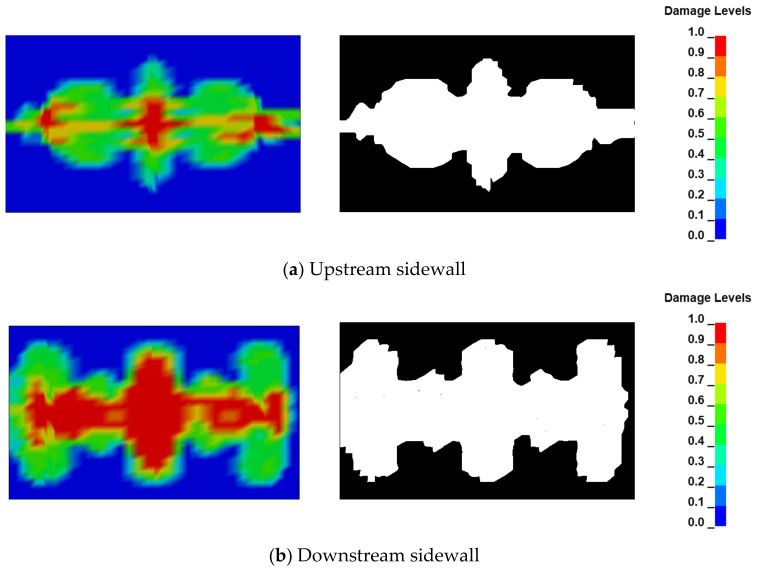
Surrounding rock damage under Case B.

**Figure 12 sensors-24-07738-f012:**
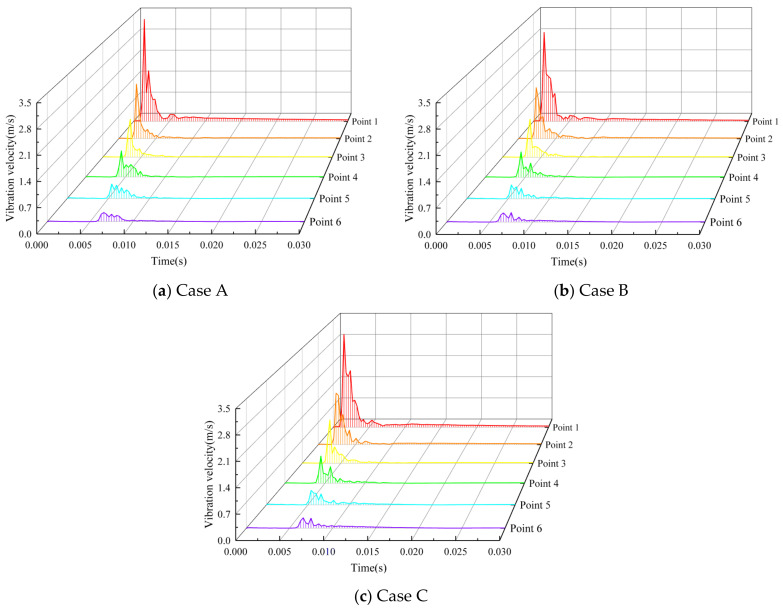
Time history curves of vibration velocity at each measuring point under three cases.

**Figure 13 sensors-24-07738-f013:**
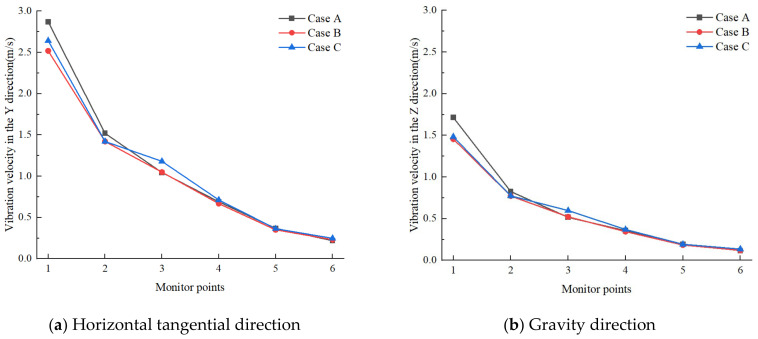
Peak vibration velocity in different directions at each measurement point.

**Figure 14 sensors-24-07738-f014:**
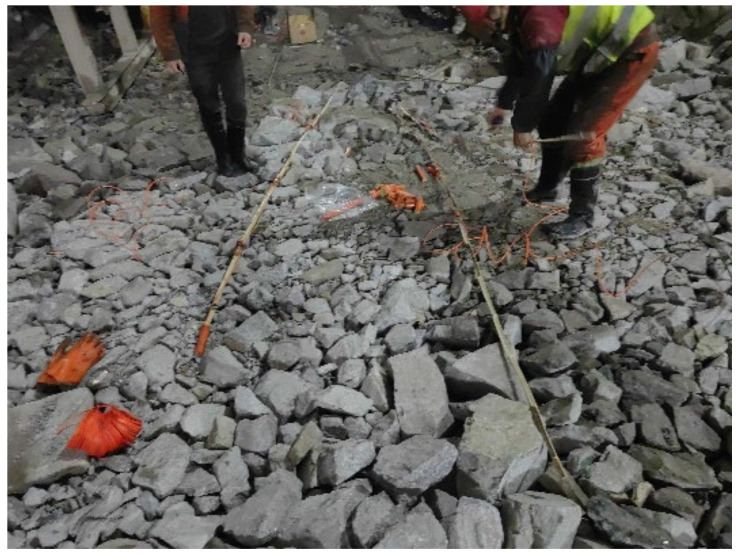
On site diagram of charge layout.

**Figure 15 sensors-24-07738-f015:**
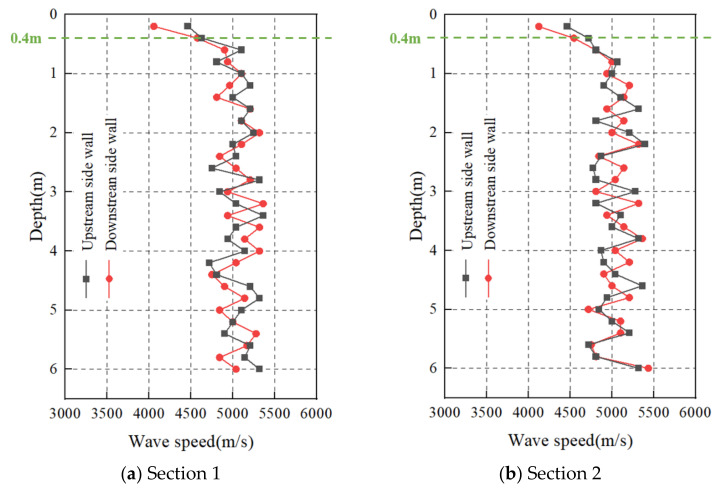
Variation in wave velocity with hole depth.

**Figure 16 sensors-24-07738-f016:**
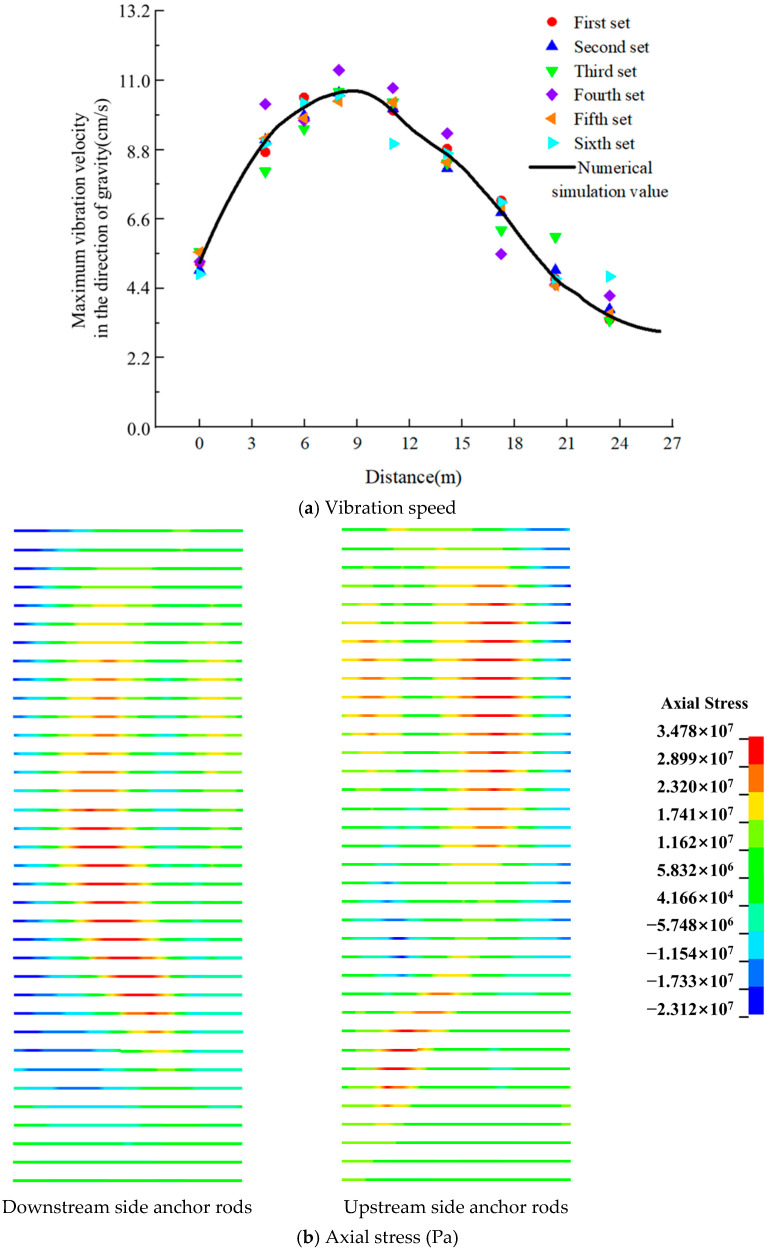
Numerical simulation values and field survey values during the blasting of the machinery room.

**Table 1 sensors-24-07738-t001:** Parameters of the RHT model for quartz monzonitic porphyry.

Parameters	Meaning	Value	Parameters	Meaning	Value
RO	Density	2730 kg/m^3^	BETAC	Compressive strain rate	0.032
SHEAR	Shear modulus	20 GPa	BETAT	Tensile strain rate	0.036
ONEMPA	/	1.0 × 10^6^	PTF	Pressure influence on plastic flow in tension	0.001
EPSF	Eroding plastic strain	2.0	GC*	Compressive yield surface	1.0
B0	Polynomial EOS	1.22	GT*	Tensile yield surface	1.7
B1	Polynomial EOS	1.22	XI	Shear modulus reduction factor	0.5
T1	Polynomial EOS	40 GP	D1	Damage parameter	0.04
A	Failure surface	2.618	D2	Damage parameter	1
N	Failure surface	0.7985	EPM	Minimum damaged residual strain	1.0 × 10^−3^
FC	Compressive strength	0.169 GPa	AF	Residual surface parameter	0.873
FS*	Relative shear strength	0.234	NF	Residual surface parameter	0.559
FT*	Relative tensile strength	0.0245	GAMMA	Gruneisen gamma	0
Q0	Load angle	0.567	A1	Hugoniot polynomial coefficient	4.0 × 10^10^
B	Load angle	0.0105	A2	Hugoniot polynomial coefficient	0
T2	Polynomial EOS	0	A3	Hugoniot polynomial coefficient	0
E0C	Ref. compressive strain rate	3.0 × 10^−5^	PEL	Crush pressure	113 MPa
E0T	Ref. tensile strain rate	3.0 × 10^−6^	PCO	Compaction pressure	5.1 GPa
EC	Break compressive strain rate	3.0 × 10^22^	NP	Porosity exponent	3
ET	Break tensile strain rate	3.0 × 10^22^	ALPHA	Initial porosity	1

**Table 2 sensors-24-07738-t002:** Proportion of damage area on sidewalls of different charge structures.

Position	Charge Structure		
	A	B	C
Upstream	0.5278	0.3840	0.3224
Downstream	0.6142	0.5403	0.4927

**Table 3 sensors-24-07738-t003:** Wave velocity from 0.4 m to the bottom of the hole (unit: m/s).

Section 1	Upstream			Downstream		
	Minimum	Average	Maximum	Minimum	Average	Maximum
	4717	5039	5362	4751	5056	5362
Section 2	Upstream			Downstream		
	Minimum	Average	Maximum	Minimum	Average	Maximum
	4717	5054	5391	4717	5076	5435

## Data Availability

The original contributions presented in the study are included in the article, further inquiries can be directed to the corresponding author.

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
