# Peer review of "Geostress-Adaptive Charge Structure Design and Field Validation for Machinery Room Excavation"

_sensors, 2024, doi:10.3390/s24237738_

Round 1
Reviewer 1 Report
Comments and Suggestions for Authors
While achieving the coupling effect of geostress and anchor rods, the rock fragmentation rate and the vibration effect induced by blasting are comprehensively considered. This article is comprehensive, logically organized and contains valuable information. Publication of those results will certainly be of practical and academic interest to many readers of this journal.
However, there are few things need to be corrected and included in the manuscript for better understanding of carried research work to the reader.
(1). How does the sensor shown in Figure 2 work? More details of this device are attractive to reader.
(2). Why are downstream parameters given first and then upstream parameters given in Figures 9, 10, and 11? Is there a theoretical basis for doing so?
(3). Most of the images in the paper are clear and concise, but for the convenience of readers to better understand, can you unify the colors in Figure 6 (b) with Figure 7, and Figure 9 with Figure 10?
(4). Overall, this work is well written, but some spelling errors can be found throughout the manuscript, such as “virbration” in figure 15 (a). This problem should be improved.
Author Response
Reviewer 1:
While achieving the coupling effect of geostress and anchor rods, the rock fragmentation rate and the vibration effect induced by blasting are comprehensively considered. This article is comprehensive, logically organized and contains valuable information. Publication of those results will certainly be of practical and academic interest to many readers of this journal.
We thank the reviewer for the positive review.
However, there are few things need to be corrected and included in the manuscript for better understanding of carried research work to the reader.
- How does the sensor shown in Figure 2 work? More details of this device are attractive to reader.
A: Thank you for the valuable comments. The sensor workflow has been added in section 2.2.1, which is highlighted in green in the revised manuscript.
- Why are downstream parameters given first and then upstream parameters given in Figures 9, 10, and 11? Is there a theoretical basis for doing so?
A: In the construction site, the downstream is on the left side and the upstream is on the right side, hence, it is customary in the manuscript to distribute from left to right. But for your convenience in understanding, corrections have been made to Figures 9, 10 and 11.
- Most of the images in the paper are clear and concise, but for the convenience of readers to better understand, can you unify the colors in Figure 6 (b) with Figure 7, and Figure 9 with Figure 10?
A: Thanks for the valuable feedback from the reviewer. The colors of Figures 6 (b) and 7, as well as Figures 9 and 10, have been unified in the revised manuscript.
- Overall, this work is well written, but some spelling errors can be found throughout the manuscript, such as “virbration” in figure 15 (a). This problem should be improved.
A: We apologize for the typing errors and the overall manuscript has been checked and revised.

Reviewer 2 Report
Comments and Suggestions for Authors
This manuscript presents a modeling approach to optimize blasting damage for excavation while limiting damage to surrounding rock masses. Three distributions of changes were compared in simulations, with a "best" choice used in a field demonstration. In place of the damage metric used in the simulations, axial stresses are used to compare the model to the field test. In a limited case study, the model closely predicted the stresses from the field test.
It is not clear how well this modeled approach would generalize to other excavation settings. I would like to see some added discussion from the authors on this topic so that others can better understand if it would be applicable to their projects and the limitations of this approach.
What was the time step used in the simulations and how long did it take to run? I wonder if the choice of time step could have an effect on the peak vibration velocities from the different Cases. Is it possible that the 3.96% difference between Cases B & C would change if the time step was different. I also suggest providing details about computing hardware used for the simulation.
In the paragraph beginning in Line 338, I don’t understand why the authors were not able to directly use the sidewall values from the simulations to calculate damage area without requiring MATLAB and grey scale image processing.
Line 400: Until I reached the conclusion, it was not clear to me which charging scheme was used for the field demonstration and why. This should be stated more clearly, using the results from Section 4.3 & 4.4. Details such as those provided in Line 479 (“Given the necessity…Case A was deemed unsuitable”) and Line 494 should be included in the main text before discussing the results from the field demonstration in Section 5.
Line 417 “In comparison to analogous projects under similar geological conditions”: please add citations for these projects.
In Lines 317-355, Table 2, and the Conclusion: it is somewhat confusing that the damage percentages and differences in these percentages are both reported as percentage values. I would suggest changing the damage percentages to damaged fractions and leaving the relative increases or decreases as percentage value (ex. from the Conclusion: “damage percentages of 43.85% and 39.19%” can become “damaged fractions of 0.4395 and 0.3919”).
Figure 6: in each diagram, is very difficult to see the colors that are listed in the legend or text (ex. charge, blockage, air). Points 1-6 are also hard to read.
In an early diagram such as Figure 6, I would suggest labeling the locations of labels used in later figures, such as “downstream”, “intermediate section”, “midsection groove”, “upstream”, “sidewalls”. This would make it much easier to switch between each of the modeling figures.
Figure 8: each column should be labeled with the time being shown (0.6 ms, 1.5 ms, and 30 ms). The default colorbar should also be adjusted, removing the scientific notation to better match the text (0.0, 0.1, ..., 1.0) and replacing the "Fringe Levels" label with text about damage.
Figure 13: I suggest using the same y-axis scale for each plot, to better communicate the lower velocities in the Z direction.
There are two figures labeled as “Figure 13” in the manuscript. Please correct and adjust all text references as needed.
Figure 15a: why does the numerical simulation curve not have a similar shape as those shown in Figure 13? Maybe I don't understand the geometry differences between the model and field test that caused velocity to peak around 7 - 11 meters.
Figure 15b: This plot seems key to the paper as it compares modeling and field results, but it is impossible to see or understand. Please increase the thickness of the plotted lines and add x and y axes to the plot. Here I also suggest changing the colorbar. If possible, there should be a clear change in color when axial stresses go from compressive to tensile.
Author Response
Reviewer 2:
This manuscript presents a modeling approach to optimize blasting damage for excavation while limiting damage to surrounding rock masses. Three distributions of changes were compared in simulations, with a "best" choice used in a field demonstration. In place of the damage metric used in the simulations, axial stresses are used to compare the model to the field test. In a limited case study, the model closely predicted the stresses from the field test.
(1). It is not clear how well this modeled approach would generalize to other excavation settings. I would like to see some added discussion from the authors on this topic so that others can better understand if it would be applicable to their projects and the limitations of this approach.
A: Thank the reviewer for the valuable comments. The limitations have be noticed and marked in Conclusion as “It should be noted that the established model is applicable only to areas with moderate to low ground stress. The rock mass size of the segment to be excavated in the project can be proportionally converted to the corresponding dimensions aforementioned. Satisfactory blasting schemes can be achieved without incurring any additional economic costs.”.
(2). What was the time step used in the simulations and how long did it take to run? I wonder if the choice of time step could have an effect on the peak vibration velocities from the different Cases. Is it possible that the 3.96% difference between Cases B & C would change if the time step was different. I also suggest providing details about computing hardware used for the simulation.
A: With the explicit finite element analysis software LS-DYNA, the blasting was simulated over a total duration of 30 milliseconds, which was divided into 100 time steps, each corresponding to a time increment of 0.03 milliseconds. The peak vibration velocity varies with the structure of the charge and the position of the measuring point, but the time step does not affect it. In addition, the hardware used for numerical simulation has been added and marked in Section 4.1.
(3). In the paragraph beginning in Line 338, I don’t understand why the authors were not able to directly use the sidewall values from the simulations to calculate damage area without requiring MATLAB and grey scale image processing.
A: Numerical simulation is based on the finite element method, so grids with damage greater than 0.6 at the sidewall can be removed and then calculated, but the accuracy of this method is not too high. Considering the need to focus on the damage caused by blasting to the sidewall in the project, we chose this surface instead of a grid group for grayscale image processing to obtain more accurate results.
(4). Line 400: Until I reached the conclusion, it was not clear to me which charging scheme was used for the field demonstration and why. This should be stated more clearly, using the results from Section 4.3 & 4.4. Details such as those provided in Line 479 (“Given the necessity…Case A was deemed unsuitable”) and Line 494 should be included in the main text before discussing the results from the field demonstration in Section 5.
A: Thank the reviewer for the valuable comments. We have made a series of supplements at the beginning of Section 5.
(5). Line 417 “In comparison to analogous projects under similar geological conditions”: please add citations for these projects.
A: Sorry for the missing citations and we have added the corresponding information in the revised manuscript.
(6). In Lines 317-355, Table 2, and the Conclusion: it is somewhat confusing that the damage percentages and differences in these percentages are both reported as percentage values. I would suggest changing the damage percentages to damaged fractions and leaving the relative increases or decreases as percentage value (ex. from the Conclusion: “damage percentages of 43.85% and 39.19%” can become “damaged fractions of 0.4395 and 0.3919”).
A: Thank the reviewer for the valuable feedback. We have made revisions and marked them in the manuscript.
(7). Figure 6: in each diagram, is very difficult to see the colors that are listed in the legend or text (ex. charge, blockage, air). Points 1-6 are also hard to read.
A: We are sorry for the inconvenience caused to you. We have made modifications to Figure 6.
(8). In an early diagram such as Figure 6, I would suggest labeling the locations of labels used in later figures, such as “downstream”, “intermediate section”, “midsection groove”, “upstream”, “sidewalls”. This would make it much easier to switch between each of the modeling figures.
A: Thanks for the valuable feedback from the reviewer. We have marked them in the designated image.
(9). Figure 8: each column should be labeled with the time being shown (0.6 ms, 1.5 ms, and 30 ms). The default colorbar should also be adjusted, removing the scientific notation to better match the text (0.0, 0.1, ..., 1.0) and replacing the "Fringe Levels" label with text about damage.
A: Modifications have been made to Figure 8.
(10). I suggest using the same y-axis scale for each plot, to better communicate the lower velocities in the Z direction.
A: Thank you for your valuable feedback. We have made modifications to Figure 13 (b).
(11). There are two figures labeled as “Figure 13” in the manuscript. Please correct and adjust all text references as needed.
A: We apologize for this error and it has been corrected in the revised manuscript.
(12). Figure 15a: Why does the numerical simulation curve not have a similar shape as those shown in Figure 13? Maybe I don't understand the geometry differences between the model and field test that caused velocity to peak around 7-11 meters.
A: The measuring points in Figure 13 are located behind the excavation section, while the measuring points in Figure 16 (a) are distributed on the rock-anchored beam above the excavation section. Due to significant differences in the placement of measurement points, the shape of the numerical simulation curve varies.
Due to rock blasting and fragmentation within the range of 0-10 m, as well as non-reflective boundary condition at 0 m, the propagation and superposition of blasting stress waves generated by multiple boreholes result in the strongest vibration response at 7-11 m.
(13). Figure 15b: This plot seems key to the paper as it compares modeling and field results, but it is impossible to see or understand. Please increase the thickness of the plotted lines and add x and y axes to the plot. Here I also suggest changing the colorbar. If possible, there should be a clear change in color when axial stresses go from compressive to tensile.
A: Thanks for the valuable feedback from the reviewer. We have increased the thickness of the lines and added x and y coordinates in Figure 16 (b). However, after multiple modifications to the color scheme, it was found that the original color scheme still had the largest span, so this point was retained.

Round 2
Reviewer 2 Report
Comments and Suggestions for Authors
The authors have sufficiently addressed the majority of my comments and suggestions. I have only two remaining comments:
I still do not understand what is being shown in Figure 16b. From close reading of the text, it appears that the X axis is showing depth and Y axis is anchor number? If so, tick labels or a scale are needed. I would also like an explanation for why the simulation and data show different patterns of tensile and compressive stress. As shown, they almost look like mirror images of each other, with the X axes flipped.
In each sentence where the damage percentages have been changed to a decimal value, the text should also be changed from "damage percentage" to "damage fraction".
Author Response
The authors have sufficiently addressed the majority of my comments and suggestions. I have only two remaining comments:
We thank the reviewer for the positive review.
- I still do not understand what is being shown in Figure 16b. From close reading of the text, it appears that the X axis is showing depth and Y axis is anchor number? If so, tick labels or a scale are needed. I would also like an explanation for why the simulation and data show different patterns of tensile and compressive stress. As shown, they almost look like mirror images of each other, with the X axes flipped.
A: Figure 16(b) presents the internal force distribution of the upper row anchor rods on both the upstream and downstream sides within the overall model depicted in Figure 4. During the numerical simulation, the anchor rods are divided into 30 elements. Some elements are subjected to tension and some are compression, due to the impact of the blasting wave and the surrounding rock.

Figure 4 Overall model of the plant section to be excavated.
- In each sentence where the damage percentages have been changed to a decimal value, the text should also be changed from "damage percentage" to "damage fraction".
A: Thank you for the valuable feedback. We have made revisions to the manuscript.
